# Infrared Small Target Detection Based on Multiscale Kurtosis Map Fusion and Optical Flow Method

**DOI:** 10.3390/s23031660

**Published:** 2023-02-02

**Authors:** Jinglin Xin, Xinxin Cao, Hu Xiao, Teng Liu, Rong Liu, Yunhong Xin

**Affiliations:** School of Physics and Information Technology, Shaanxi Normal University, Xi’an 710119, China

**Keywords:** infrared small target, target detect, multiscale kurtosis map, optical flow field

## Abstract

The uncertainty of target sizes and the complexity of backgrounds are the main reasons for the poor detection performance of small infrared targets. Focusing on this issue, this paper presents a robust and accurate algorithm that combines multiscale kurtosis map fusion and the optical flow method for the detection of small infrared targets in complex natural scenes. The paper has made three main contributions: First, it proposes a structure for infrared small target detection technology based on multiscale kurtosis maps and optical flow fields, which can well represent the shape, size and motion information of the target and is advantageous to the enhancement of the target and the suppression of the background. Second, a strategy of multi-scale kurtosis map fusion is presented to match the shape and the size of the small target, which can effectively enhance small targets with different sizes as well as suppress the highlighted noise points and the residual background edges. During the fusion process, a novel weighting mechanism is proposed to fuse different scale kurtosis maps, by means of which the scale that matches the true target is effectively enhanced. Third, an improved optical flow method is utilized to further suppress the nontarget residual clutter that cannot be completely removed by multiscale kurtosis map fusion. By means of the scale confidence parameter obtained during the multiscale kurtosis map fusion step, the optical flow method can select the optimal neighborhood that matches best to the target size and shape, which can effectively improve the integrity of the detection target and the ability to suppress residual clutter. As a result, the proposed method achieves a superior performance. Experimental results on eleven typical complex infrared natural scenes show that, compared with seven state-of-the-art methods, the presented method outperforms in terms of subjective visual effect, as well as some main objective evaluation indicators such as BSF, SCRG and ROC, etc.

## 1. Introduction

The infrared imaging system is widely used in military and civil fields [1]. Due to the long imaging distance [2], infrared images obtained by thermal radiation are characteristic as follows: (1) poor resolution; (2) low contrast; (3) low signal-to-noise ratio; (4) the target usually occupies a few to a dozen pixels; (5) there is no linear relationship between gray distribution and target reflection characteristics [3,4,5,6]. Moreover, complex factors such as buildings, trees and rivers often appear in the realistic background, which are present in IR images in the forms of highlighted backgrounds, strong edges and other forms of noise that interfere with the target [7]. As a result, it is a great challenge to detect small infrared targets in complex backgrounds.

Figure 1 shows a typical infrared small target image in a complex background along with the 3D maps of its various local components, where TT, NB, HB, EB and PNHB stand for true target, normal background, high-brightness background, edge of background and pixel-sized noise with high brightness, respectively. It can be seen from Figure 1 that the real target does not exceed 6 × 6 pixels and is approximately Gaussian distributed, while the distribution of PNHB is very similar to the real target distribution but usually occupies only one pixel. How to eliminate PNHB and other complex noise interference through these differences and to achieve a high detection rate and a low false alarm rate is the final purpose of the infrared target detection field. 

So far, researchers have proposed many effective methods based on the imaging characteristics of infrared images. The existing algorithms can be roughly classified into three broad categories: single-frame detection, multi-frame (or sequence) detection [8] and the detection method of deep learning [9]. 

### 1.1. Single-Frame Infrared Small Target Detection

Single-frame detection refers to detecting targets in one image frame. It has two basic methods, namely background suppression and target enhancement, based on the significant differences between the target pixel points and the background pixel points in the image. The typical methods are the filter-based detection method, the morphology-based detection method, the human visual system (HVS) method, the low-rank sparse detection method and the gray distribution-based methods.

#### 1.1.1. Filter-Based Detection Method

In the early stage, a series of traditional methods, such as maximum mean filter [10], maximum median filter [11], bilateral filter [12] and so on, are mainly used to suppress the background, which is simple in principle and easy to implement. However, when the background is complex, the detection results will be inaccurate. In the subsequent research, researchers made many improvements based on traditional filtering methods, such as Cao et al. [13], who proposed a two-dimensional minimum mean square (TDLMS) filter to adapt to more complex backgrounds, and Zhang et al. [14], who proposed a two-layer TDLMS filter method to simultaneously extract the target and suppress the background, etc. These improved methods are highly adaptable to complex environments compared with traditional methods, but false alarms occur in the detection of locations with large changes in grayscale such as background edges, pixel-sized highlighting noise, etc.

#### 1.1.2. Morphology-Based Detection Methods

As a nonlinear filtering method, the morphological method can highlight the target and suppress the background and noise by setting a specific shape window to traverse the image and by performing basic operations such as erosion and expansion at local pixel points [15]. Kim et al. [16] performed target detection by Laplacian Gaussian (LoG) scale space theory; however, it is computationally intensive. Bai et al. [17] improved the traditional top-hat algorithm by dividing the original structured window into two layers, capturing small targets by the central window and acquiring background regions by the edge layer to achieve target detection, which has good performance for relatively simple background suppression. Liu et al. [18] added the locally significant algorithm to the improved top-hat double-layer window, etc. However, morphology-based detection methods are necessary to consider the matching process between the window shape and size and that of the real target [19]; however, in the practical detection process, the target’s shape and size are often unknown, thus leading to the problems of missed detections and false alarms [20].

#### 1.1.3. Human Visual System (HVS) Methods

The local contrast method is proposed based on the fact that the human eye is more sensitive to the contrast information in vision [21,22,23]. Based on saliency features, Wang et al. [24] proposed a differential of Gaussian filter (DoG). Later, Han et al. [25] proposed an improved difference of Gabor (IDoGb) filter. Due to the filter being sensitive to edges, it is easy for the detection result to generate a high false-alarm rate. Chen et al. [26] proposed a local contrast measure (LCM) that uses two nested windows to enhance the target contrast by capturing the target and background separately and achieving a high detection rate; it has attracted wide attention in the field of infrared small targets since its proposal [27]. Subsequently, a series of improved algorithms based on the LCM algorithm have been proposed, such as the improved local contrast algorithm (ILCM) [28], the new local contrast algorithm (NLCM) [29], etc. In the latest study, researchers have focused on the problem of unknown target size. Wei et al. [30] proposed the multiscale block contrast method (MPCM), Han et al. [3] proposed the relative local contrast algorithm (RLCM), Deng et al. [31] proposed the entropy-based weighted local contrast algorithm (WLDM), etc. The matching problem of the target is achieved by multiscale computation, but it greatly increases the computational effort.

#### 1.1.4. Low-Rank Sparse Detection Method

Using the theory of nonlocal self-correlation of background, the target and the background are considered as sparse matrix and low-rank matrix, respectively, while small target detection is realized by the mathematical optimization of low-rank matrix and sparse matrix. Gao et al. [32] proposed an IPI model to achieve more accurate image segmentation and small target extraction by optimizing the process of infrared image reconstruction. When the complexity of the background of the detected image is increased, it will be found that the IPI model is insufficient to observe the sparsity between patches, resulting in the sparse background noise also being incorrectly decomposed into the sparse matrix, interfering with the detection of real targets. Later, Dai et al. [33] introduced additional non-negative constraints and singular value partial minimization to enhance the background suppression capability of the IPI model. Unfortunately, it also suffers from the drawback of inaccurate estimation of the background matrix in the face of strong edge structures.

#### 1.1.5. Gray Distribution-Based Methods

The kurtosis map method is based on the fact that the shape of the gray distribution of an infrared small target is similar to that of the Gaussian distribution [34]. The kurtosis operation can not only enhance the target but also effectively suppresses the complex background. However, when faced with a real scene where the target size is unknown, it easily leads to target missed or false positives. Wang et al. [35] proposed a multiscale kurtosis map fusion method to detect small targets, which can effectively match the target size and shape by selecting multiscale windows and achieves high accuracy and adaptable detection for infrared small targets. However, when encountering pixel-sized noises with high brightness, false detection will still occur.

### 1.2. Multiframe Infrared Small Target Detection

The multiframe (temporal) method can use both spatial and temporal information for target detection; it refers to the effective segmentation of target and background information based on strong correlation between frames, which can be classified into the detection before tracking (DBT) method and the tracking before detection (TBD) method [36].

The DBT method refers to obtaining the initial detection target by a single-frame detection algorithm and then acquiring the real target by the continuity of target motion between consecutive frames. There are many classical methods, such as the Markov random fields method [37], the energy accumulation method [38], the Kalman filter [39], the particle filter [40], etc. These algorithms are simple in idea and easy to implement, but the subsequent correlation verification has strong requirements on the detection performance of the previous frame.

The TBD method extracts targets based on the feature that the background in the sequence images is relatively stable and only the target pixels are changing in the detection procedure. The pipeline filter [41], the 3D matched filter [42] and the 3D DoG filter [43] are three early sequence detection algorithms. These methods are only effective in specific scenes and cannot achieve good detection results in complex scenes. The interframe difference type method is a common sequence target detection technology, such as adjacent frame difference [44], the difference in five consecutive frames method [45], etc. This type of method is simple to implement and has strong adaptive ability. However, its computational complexity is very high. The background subtraction method is another sequence target detection technology, such as the spatiotemporal background model method [46], the coding model background subtraction method [47], etc. These algorithms are simple in principle and fast in detection; however, when the actual background changes, the accuracy of the detection results will also change.

In addition, the optical flow method extracts the target by using the moving characteristics of the target and the static characteristics of the background in the image sequence. The optical flow field methods include local calculation and global calculation. The L-K (Lucas–Kanade) method [48] is the classical local calculation method and the H-S (Horn–Schunck) method is the classical global calculation method. LK optical flow only calculates the optical flow of sparse points [49], which can greatly improve the operation speed. The optical flow method can be used in target detection without knowing any scene information in advance, but factors such as noise, multiple light sources and shadows can affect the calculation of the optical flow field distribution and thus affect the target detection results.

### 1.3. The Detection Method of Deep Learning

With the continuous improvement of computer computing power, a large number of target detection algorithms based on deep learning have been proposed in recent years. Among them, the candidate frame-based weak and small target detection method R-CNN [50] first combines the candidate frame with the convolutional neural network, which can reduce the computational complexity and improve the detection accuracy, but it is easy to cause image distortion and target loss when the number of candidate frames is too large and a fixed size is required. Joseph Redmon et al. [51] proposed the YOLO algorithm for the problems of complicated steps and slow training speed of the candidate frame series algorithm, which directly divides the input image, eliminating the process of searching the image and generating candidate frames, and truly realizes an end-to-end target detection and the detection speed can reach the standard of real-time processing. However, because it lacks the search process for candidate frames, the algorithm is unable to accurately locate the target and there is much room for improvement in detection accuracy and recall. In 2021, Song et al. [9] proposed a novel network, AVILNet, following the latest methods GridNet [52], Cross-Stage-Partial-Net [53] and atrous convolution [54] among others composed to solve infrared small target segmentation, successfully coordinating the balance between recall and accuracy. So far, the infrared dim and small image detection algorithm based on deep learning is still in the emerging stage; however, the data set is seriously deficient and there is still huge room for improvement in the detection performance of infrared dim and small targets.

### 1.4. Motivation

As mentioned above, the current research mainly focuses on single-frame detection. When the detection background is extremely complex, such as a large number of highlight noises and background edges, there would be some false alarms using single-frame detection. On the other hand, the optical flow method can detect the target by taking advantage of its moving nature; however, it is sensitive to noise. Considering the features of these two types of methods, we have presented an effective infrared small target detection based on multiscale kurtosis map fusion and the optical flow method, which makes full use of the information of the target shape and size and motion characteristics for its detection and achieves excellent performance.

## 2. Proposed Method

The framework of our proposed method is shown in Figure 2. It can be seen that the method consists of four steps: preprocessing, multiscale kurtosis map fusion, optical flow method and target extraction. In the preprocessing step, a Gaussian high pass filter is used to remove the large-area uniform background, which is a routine procedure in image processing. The step of multiscale kurtosis map fusion is applied to match the shape and the size of the small target, which can effectively enhance small targets with different sizes, as well as suppress the highlighted noise points and the residual background edges. The optical flow step is utilized to further suppress the nontarget residual clutter that cannot be completely removed by multiscale kurtosis map fusion. The target extraction step is a simple one and, by means of simple operation, i.e., threshold segmentation and threshold extraction, the final target image can be obtained.

### 2.1. Preprocessing

The aim of the image preprocessing is to remove the large-area uniform background, meanwhile retaining the target information as much as possible. Small targets usually occupy only a few to dozens of pixels and always present high-frequency characteristics. Therefore, here we use Gaussian high-pass filtering to achieve this task. Gaussian filtering is a kind of linear smoothing filtering and is widely used in the noise reduction process of image processing. Due to the gray distribution of small targets being close to the Gaussian distribution, the use of the Gaussian filter has an additional advantage, that is, it can make somewhat of an enhancement for the target.

The two-dimensional Gaussian high-pass filter function can be expressed as follows:(1)$$G\left(x,y\right)=1-e^{-\left(\frac{D^{2}\left(x,y\right)}{2D_{0}{}^{2}}\right)\;\;\;}\;$$
where coordinates *(x*, *y)* represent the position of the pixel, $D_{0}\;$ characterizes the cutoff frequency and *D (x*, *y)* is the distance from the frequency center. Using this function, it can generate a filter template and the filtered image can be obtained by convolving the template and the image.

Figure 3 shows the result of the Gaussian high-pass filter, where the large-area uniform backgrounds are removed and the target pixels are highlighted, but some HBE and PNHB remain.

### 2.2. Multiscale Kurtosis Map Fusion

#### 2.2.1. Kurtosis and Kurtosis Map

The Gaussian high-pass filter preprocessing step is a rough process procedure; there is still some clutter, such as HBE and PNHB, in the image after preprocessing. Consequently, a multiscale kurtosis map fusion method is proposed to highlight the target and to suppress the residual clutter. Kurtosis is also called the kurtosis coefficient; it is a characteristic number representing the peak value of the probability density distribution curve at the average value. Compared with the normal distribution, if the kurtosis is greater than three, the peak shape is sharp and steeper than the normal distribution peak and vice versa. For an image, the kurtosis can reflect the similarity between the distribution of gray values centered on the window and the two-dimension Gaussian distribution.

The kurtosis *K* of a random variable *X* is defined as the ratio of its fourth central moment to the square of its variance.
(2)$$K=\frac{E{\left(x-\mu \right)}^{4}}{\sigma^{4}}\;$$
where *μ* is the mean value of random variable *X* and $\sigma^{2}$ is its variance. In practical applications, the originally defined kurtosis is usually subtracted by three so that its zero value corresponds to the Gaussian distribution; this new parameter is still called kurtosis.

For a pixel of an image, its kurtosis is defined on its local adjacency area and the definition express is as follows,
(3)$$K\left(i,j\right)=\frac{1}{N_{w}}\frac{\sum_{\left(x,y\right)\epsilon w\left(i,j\right)}{\left(I\left(x,y\right)-\mu \right)}^{4}}{\sigma^{4}}-3$$
where $w\left(i,j\right)$ denotes the window centered on the pixel $\left(i,j\right)$*,* $N_{w}$ denotes the number of pixels in the window $w\left(i,j\right)$ and *μ* and *σ* denote the mean and standard deviation of the gray value of the pixels in $w\left(i,j\right)$, respectively.

It should be noted that the kurtosis value is defined in a specific local area, i.e., the window $w\left(i,j\right)$, thus this value is decided by the shape and size of the selected window $w\left(i,j\right)$; that is to say, the selection of window is extremely important for the kurtosis.

When the window is selected, moving the window from left to right and from top to bottom along an image and calculating the kurtosis value of each point can obtain the kurtosis map of the image. Algorithm 1 gives the computational steps of the kurtosis map.
**Algorithm 1** The Computational Steps of the Kurtosis Map
Input: initial infrared image
I,  size  M×N; window  W, size  U×V 
Preprocessing: padding $\left(U-1\right)/2$ zero points on the left and right of the image I  and padding $\left(V-1\right)/2$ zero points on the top and bottom of the image *I*;For $i=\left(U-1\right)/2\;$: $M+\left(U-1\right)/2$; for $\left(V-1\right)/2$: $N+\left(V-1\right)/2$ do;Calculate the mean  μ and the variance $\;\sigma^{2}$ of the pixels in $w\left(i,j\right)$;$k=i-\left(U-1\right)/2;$$l=j-\left(V-1\right)/2$$\;;\;K\left(k,l\right)=0;$For $\;x=-\left(U-1\right)/2$$:\;\left(U-1\right)/2$; for $\;y=-\left(V-1\right)/2$$:\;\left(V-1\right)/2$ do;$K\left(k,l\right)=K\left(k,l\right)+\frac{{\left(I\left(i+x,j+y\right)-\mu \right)}^{4}}{U\times V\times \sigma^{4}}-3;$End for *x*; end for *y*;End for *i*; end for *j*.Output: kurtosis map *K*, size M×N


Figure 4 shows three kurtosis maps obtained with 3 × 3, 5 × 5 and 7 × 7 windows, respectively. It can be intuitively seen that in the kurtosis maps the target area (inside the red circle) and the highlight noise area (inside the blue circle) is enhanced brightly. With a different window selected, the size of the white area is also different and with the increase of the window, the white area also increases; meanwhile, the differentiation of the background edge increases. When the size of the selected window is matched with that of the target, such as with size 3 × 3, the result for target enhancement and clutter suppression is perfect (see Figure 4a).

#### 2.2.2. Multiscale Kurtosis Map Fusion

The size and shape of the target cannot be known in advance. In order to improve the adaptability of the algorithm to different target sizes, we present a novel multiscale kurtosis map method to achieve this work, in which the scale corresponds to the size of the window and the multiscale kurtosis maps refer to that obtained by different sizes of windows. The key technology of the multiscale kurtosis map method is the strategy of the multiscale fusion. After theoretical analysis and experimental verification, we have developed a high-performance multiscale fusion technique named the max-min weighted fusion method; its principle is as follows.

In order to facilitate subsequent calculation processes, the shapes of all the windows are set as square and the lengths of their side are odd. The advantage of this is that the window has only one parameter and the calculation area is symmetrical around the pixel. 

Given that there are S scales, i.e., S windows with the size $W_{1}$, $W_{2}$,⋯, $W_{s}$, their corresponding kurtosis maps are $K_{1}$, $K_{2}$, ⋯, $K_{s}$, respectively. Theoretically, if the influence of noise were ignored, the kurtosis map generated by the target would be symmetrical. In other words, the kurtosis values of the pixels at the edge of the window should be approximately equal. Therefore, this feature can be used as a discrimination factor for target pixels, considering the fact that the kurtosis values of uniform background edges are also approximately equal as well as the kurtosis feature of the target pixel. To distinguish the target from the background, we have defined a confidence parameter $b_{s}\left(i,j\right)$ that denotes the possibility of a pixel being a target pixel; it is described as follows:(4)$$b_{s}=\frac{K_{s}\left(i,j\right)}{1+Var\left\{edge\;pixels\;except\;four\;corners\;of\;w_{s}\left(i,j\right)\right\}},\;i=1,\cdots ,M;j=1,\cdots ,N\;$$
where $K_{s}\left(i,j\right)$ denotes the s-th kurtosis values of the pixel $\left(i,j\right)$, $w_{s}\left(i,j\right)$ is the s-th scale window centered on the pixel $\left(i,j\right)$ of the kurtosis map $K_{s}$, 1 in the denominator is to prevent the denominator from being zero and $Var\left\{\cdot \right\}$ denotes the variance of the involved pixels.

The higher the value of $b_{s}\left(i,j\right)$, the greater the possibility that the pixel $K_{s}\left(i,j\right)\;$ belongs to a target. Moreover, $b_{s}\left(i,j\right)\;$ is also an indicator of the matching degree between the kurtosis map and the target size and it can be used as a basic parameter for multiscale kurtosis map fusion. Equation (5) is the fusion formula of the S scale kurtosis map.
(5)$$K\left(i,j\right)=\overset{S}{\underset{s=1}{\sum }}\frac{b_{s}\left(i,j\right)}{\sum_{i=1}^{S}b_{i}\left(i,j\right)}K_{s}\left(i,j\right),\;i=1,\cdots ,M;j=1,\cdots ,N$$

$K\left(i,j\right)\;$ is the final result of the kurtosis map. However, the kurtosis map only highlights the target area and removes the background residue. To obtain the target detection image, it must be combined with the infrared image. Here, we use point multiply operation to complete this task. Algorithm 2 gives the pseudocode of multiscale kurtosis map fusion.
**Algorithm 2** Multiscale Kurtosis Map FusionInput: infrared image I, size M×N; Multi-scale kurtosis map $K_{1,}K_{2},\cdots ,K_{S},\;\mathrm{size}$ M×N; For i = 1:M do;For j = 1: N do;  Calculate confidence parameter $b_{s}\left(i,j\right)$, Formula (4);  Fusion multiscale kurtosis maps $K\left(i,j\right)$, Formula (5);  Dot multiply to obtain output image $F\left(i,j\right)=G\left(i,j\right)\cdot \times K\left(i,j\right);$ End for;  End for; Output: coarse target image of multiscale kurtosis fusion $F\left(i,j\right)$.

Figure 5 shows the fusion process and the result of three scale kurtosis maps with window 3×3, 5×5 and 7×7. It can be intuitively seen that in the 3-scale fusion results, the target (inside the red circle) is accurately detected, while the background clutter is suppressed very cleanly, but the pixel-sized noise with high brightness is still retained (inside the blue circle).

### 2.3. Optical Flow Method

The optical flow method mainly detects the target by calculating the optical flow field in the scene. It does not need to know the scene information in advance and is widely used in pattern recognition, computer vision and other image processing fields. However, the optical flow field is very sensitive to the environment and the noise in the scene, multilight source human shadow and occlusion will seriously affect the distribution calculation of the optical flow field, reducing the calculation accuracy. Fortunately, the infrared image obtained by multiscale fusion has removed various environmental noises and maintained the stability of pixel brightness, which can well meet this requirement.

The main task of the optical flow method is to calculate the optical flow field, i.e., to estimate the motion field according to the spatiotemporal gradient of the image sequence under the appropriate smoothness constraints and thereafter detecting and segmenting the moving objects and scenes by analyzing the changes of the motion field. The most classical global optical flow field methods are the L-K (Lucas–Kanade) method and the H-S (Horn–Schunck) method. In this paper, we apply the L-K method to calculate the optical flow field of an infrared image. The L-K algorithm is based on three fundamental assumptions.

Assumption 1: the pixel intensities do not change between consecutive frames.Assumption 2: neighboring pixels have a motion similar to that of the central pixel.Assumption 3: the pixels have subpixel motion, which means they move less than 1 pixel between two consecutive frames.

According to the basic assumptions of the optical flow algorithm, let the small offset of the target in the x-direction be u and the small offset in the y-direction be v, then the following equation can be obtained according to the luminance consistency assumption:(6)$$I\left(x,y,t\right)=I\left(x+u,y+v,t+1\right)$$

Optical flow represents the correlation of pixels in two consecutive frames of image motion. The first order expansion of Equation (6) by Taylor’s formula can be obtained,
(7)$$I\left(x,y,t\right)=I\left(x,y,t+1\right)+\frac{\partial I}{\partial x}u+\frac{\partial I}{\partial y}v+o\left(x,y\right)$$

The sequential infrared images of small shifts ensure:(8)$$I_{t}+I_{x}\;u+I_{y}v=0$$
where $I_{t}$ is the gradient in time, $I_{X}$ and $I_{y}$ are the gradients in the space of the x and y coordinates, respectively.

The fundamental Assumption 2 of the Lucas–Kanade algorithm implies a spatial consistency constraint, that is, the neighboring points in a same plane have the same motion. Let the neighborhood of a pixel be a patch of size N×N; according to this constraint all the pixels in the patches with the same speed can be solved by the following equation.
(9)$$\left\{\begin{array}{c} I_{X}\left(P_{1}\right)u+I_{y}\left(P_{1}\right)v=-I_{t}\left(P_{1}\right)\\ I_{x}\left(P_{2}\right)u+I_{y}\left(P_{2}\right)v=-I_{t}\left(P_{1}\right)\\ \vdots \\ I_{x}\left(P_{N\times N}\right)u+I_{y}\left(P_{N\times N}\right)v=-I_{t}\left(P_{N\times N}\right) \end{array}\right.$$
where $P_{i}\left(i=1,2,\cdots ,N\times N\right)$ is the coordinate of pixel i in the selected neighborhood patch. For convenience of description and calculation, Equation (9) is usually written in matrix form.
(10)$$\left[\begin{array}{c} I_{x}\left(P_{1}\right)\;\;\;I_{y}\left(P_{1}\right)\\ I_{x}\left(P_{2}\right)\;\;\;I_{y}\left(P_{2}\right)\\ \vdots \;\;\;\;\;\;\;\;\;\;\;\;\;\vdots \\ I_{x}\left(P_{N\times N}\right)\;\;\;I_{y}\left(P_{N\times N}\right) \end{array}\right]\left[\begin{array}{c} u\\ v \end{array}\right]=-\left[\begin{array}{c} I_{t}\left(P_{1}\right)\\ I_{t}\left(P_{2}\right)\\ \vdots \\ I_{t}\left(P_{N\times N}\right) \end{array}\right]$$

Figure 6 shows the result of the object motion detection in a color image sequence with the LK optical flow method. The original sequence is a clear RBG color image in which the objects and the backgrounds are also quite clear. As a result, the optical flow field of the objects can be obviously distinguished from that of the backgrounds and the several moving objects are effectively detected.

When the optical flow method is directly used for infrared small target detection, the situation becomes more complicated. (In this paper, the optical flow algorithm selects pictures with continuous frames. In order to ensure the effectiveness of the optical flow method, the selected picture is that with the target moving in the continuous frame.) Figure 7 shows the detection result of an infrared small moving target in an infrared image under a complex background with the LK optical flow method. It can be seen that, without the kurtosis map fusion step, the optical flow detection is poorly robust for complex backgrounds and the target is too small to distinguish from the noise. Therefore, it can be concluded that it cannot effectively detect a target by directly utilizing the optical flow method in the original infrared sequence.

A benefit of the of multiscale kurtosis map fusion method is that a large number of disordered complex backgrounds are filtered out and the target is enhanced, which makes a favorable condition for subsequent detection with the optical flow method. In the output of the multiscale kurtosis map fusion step, a coarse detection image can be obtained with a small amount of clutter and enhanced targets. These clutters are difficult to be completely removed by conventional methods but, by using the moving characteristics of the target and the static characteristics of the clutter, we can effectively remove these clutters by means of the optical flow method to achieve a reliable target detection. Figure 8 shows the detection result with the optical flow method for the output image sequence of kurtosis map fusion, in which all clutter is removed and the target is effectively detected.

When applying the optical flow method, one factor that needs to be emphasized is that, similar to the calculation of kurtosis, the calculation of optical flow is also based on a neighborhood range. The size of this area determines the calculation results of optical flow; thus, the setting of the neighborhood size is also extremely important for the optical flow method as it can ultimately determine the detection performance of the target. 

Consider that in the fusion process of multiscale kurtosis maps we have calculated the confidence of the scale, which can be directly applied to the selection of the neighborhood range of the optical flow method. By this means, it not only improves the performance of the optical flow method but also improves its computational efficiency. The specific operation is that, when calculating the optical flow, the nearest neighborhood of pixel $\left(i,j\right)$ is the patch corresponding to the maximum confidence value $\;b_{s}\left(i,j\right)$. It can be expressed as follows:(11)$$w_{opt}\left(i,j\right)=w_{s}\left(i,j\right),\;max_{s}\left\{b_{s}\left(i,j\right),s=1,2,\cdots ,S\right\}$$

After determining the optimal neighborhood $w_{opt}\left(i,j\right)\;$ of a pixel $\left(i,j\right)$, an effective optical flow field image can be obtained through Formula (11), which is denoted as $O\left(i,j\right)$.

### 2.4. Target Extraction

The optical flow field is a means to exhibit the target motion state, which aims to highlight the target and eliminate static clutter by using the target motion characteristics. Utilizing the optical flow field image $O\left(i,j\right)$, a mask image $M\left(i,j\right)$ can be generated by an adaptive threshold operation. The threshold Th is defined as
(12)Th=μ+k×δ 
where *μ* and δ are the mean value and standard deviation of the image $O\left(i,j\right)$ and k is a weighting factor, which is generally selected from experiments. In the experiment, k  is set as 1.

The mask image $M\left(i,j\right)$ is a binary image that is generated by the formula (13)
(13)$$M\left(i,j\right)=\left\{\begin{array}{c} 1\;\;\;\;O\left(i,j\right)>Th\\ 0\;\;\;otherwise\;\;\; \end{array},\;\;\right.\;i=1,2,\cdots M;j=1,2,\cdots ,N\;$$

Finally, the output image $Z\left(i,j\right)$ is obtained by a simple dot multiply operation.
(14)$$Z\left(i,j\right)=M\left(i,j\right)\cdot \times R\left(i,j\right),\;i=1,2,\cdots ,M;j=1,2,\cdots ,N\;$$
where $R\left(i,j\right)$ is the output image of the preprocessing step.

## 3. Experiments and Analysis

In this section, we set up experiments on real infrared scenes that involve various complex backgrounds and compare our method with seven state-of-the-art algorithms such as ADMD [55], IPI [32], NIPPS [33], MPCM [30], NSM [56], RLCM [3] and TGDS [57] in a subjective visual and objective evaluation to comprehensively verify its effectiveness and robustness. It should be noted that some parameters of the baseline methods have been adjusted in the experiments because the default parameters set by the authors may not achieve the best performance when facing extremely complex backgrounds.

### 3.1. Datasets

The datasets consisted of eleven image sequences and the detailed characteristics of the target and the background are listed in Table 1. The scenes in these images include clouds, trees, grasslands, buildings, mountains and other complex backgrounds, as well as multiple targets, which can effectively verify the performance of the detection algorithm.

### 3.2. Evaluation Indicators

Signal-to-clutter ratio gain (SCRG) and background suppression factor (BSF) are widely used as the most important metrics of detection performance, which represent the target saliency and background suppression ability of an algorithm, respectively. They are defined as follows:(15)$$\mathrm{SCRG}=20\mathrm{lg}\left(\frac{{\left(S/C\right)}_{out}}{{\left(S/C\right)}_{in}}\right)\;$$
(16)$$\mathrm{BSF}=20\mathrm{lg}\left(\frac{C_{in}}{C_{out}}\right)\;$$
where *S* is the gray level average of the small target, *C* denotes the standard deviation of the background clutter and ${\left(\;\right)}_{in}$ and ${\left(\;\right)}_{out}$ represent the parameters of the input and the output image, respectively. The higher the SCRG and BSF values are, the superior the performance of the target enhancement and background suppression is.

The relationship between the probability of detection $\left(P_{d}\right)$ and the probability of false alarm $\;\left(P_{f}\right)\;$ can be reflected by the receiver operating characteristic (ROC) curves, which is another most important metric of detection performance. The definitions of $P_{d}$ and $P_{f}$ are as follows:(17)$$P_{d}=\frac{N_{p}}{N_{r}}\;$$
(18)$$P_{f}=\frac{N_{f}}{N_{n}}\;$$
where $N_{p}$ is the number of real pixels detected, $N_{r}$ is the number of pixels occupied by small targets, $N_{f}$ is the number of false pixels detected and $N_{n}$ is the number of pixels in the entire image. The ROC curve is obtained with the false alarm rate $P_{f}$ as the horizontal axis and the detection rate $P_{d}$ as the vertical axis.

### 3.3. Experimental Results and Analysis

Figure 9 shows the target detection images and the 3D maps of the seven baseline algorithms and our presented method, where we mark the target with a red rectangle in the images and place the 3D maps at the bottom of the images. Figure 9 vividly demonstrates the detection performance of various algorithms. In the scenarios a and b, which have cloud backgrounds, the target is not detected by the algorithms NSM and TDGS. In scenarios e and f, which contain multiple targets, the algorithms ADMD and NSM do not detect the targets completely. When the scene is extremely complex, as shown in Figure 9g–k, the NSM algorithms cannot detect the target. The results of the IPI, LCM, MPCM, RLCM and TDGS algorithms still maintain some noise. The results of different algorithms show that strong edges and bright backgrounds are the important reasons that affect the detection results. Sometimes, the desired effect cannot be achieved only by enhancing the target and suppressing the background. By means of the target-moving characteristics, the target can be distinguished from the background with the same brightness. Therefore, it can be seen that, under different circumstances, our algorithm can achieve the perfect detection.

SCRG and BSF are used to quantitatively evaluate the proposed model and baseline methods, which are shown in Table 2 and Table 3, and the best two results are highlighted with red font and blue font, respectively. The larger the value, the better the detection performance. It can be seen from Table 2 that our method achieves the best results (except for the detection results of algorithm IPI in scenario d), which indicates that our algorithm has obvious advantages in terms of the target enhancement. In Table 3, the BSF values of the NSM algorithm in scenario d, the MPCM algorithm in scenario j and the IPI algorithm in scenario k are 1.41, 1.73 and 0.12 higher than that of our method, respectively. However, these algorithms only have the highest BSF value in an individual scenario; their performance in other scenarios is not as good as our methods. Therefore, in general, our algorithm performance is better and more stable.

The ROC curves of all the detection results are shown in Figure 10. It is seen that, as the image structure becomes more and more complex, the performance of the comparison algorithms varies greatly. However, our proposed method always achieves the highest detection rate while forming the curve with the largest area enclosed by the horizontal coordinate, which indicates that our algorithm has good detection performance in various complex scenarios.

## 4. Conclusions

This paper presents a novel method that combines multiscale kurtosis map fusion and the optical flow method to improve the detection accuracy and robustness of the small infrared target in complex natural scenes, in which the multiscale kurtosis map can effectively match the size and the shape of the target and the optical flow method can well represent the motion information of the target. By these means, the presented method is advantageous to the enhancement of the target and the suppression of the background. Extensive experimental results show that, compared with state-of-the-art algorithms, the proposed method achieves promising detection results in multiple evaluation indicators and shows better stability and effectiveness when facing various complex scenarios.

## Figures and Tables

**Figure 1 sensors-23-01660-f001:**
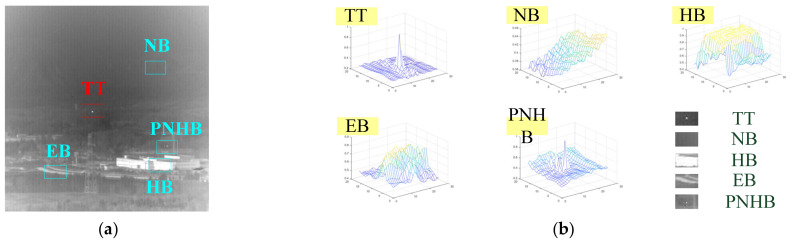
(**a**) Infrared small target image; (**b**) 3D maps of various local components of an infrared image.

**Figure 2 sensors-23-01660-f002:**
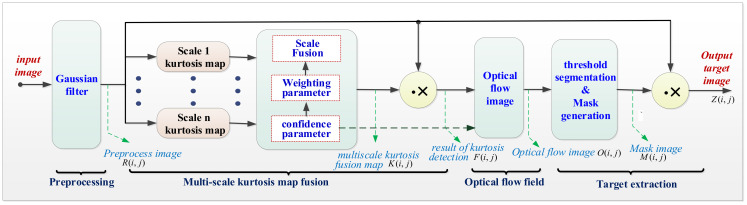
The framework of the presented method.

**Figure 3 sensors-23-01660-f003:**
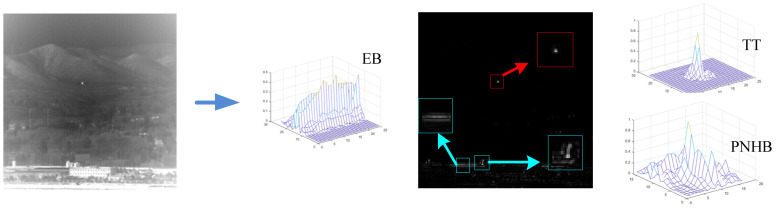
Result of the Gaussian high-pass filter.

**Figure 4 sensors-23-01660-f004:**
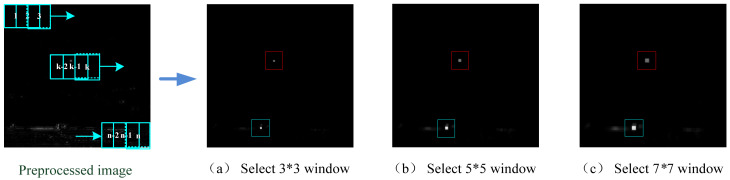
Kurtosis maps calculated by different window.

**Figure 5 sensors-23-01660-f005:**
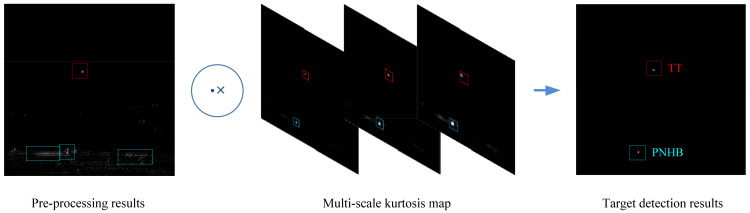
The frame of multiscale kurtosis fusion.

**Figure 6 sensors-23-01660-f006:**
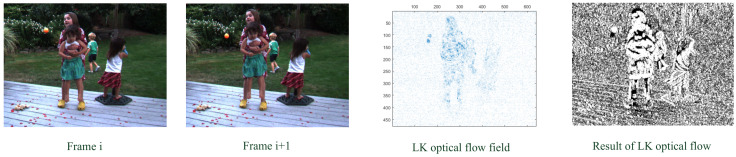
The object motion detection in color image sequence with the LK optical flow method.

**Figure 7 sensors-23-01660-f007:**
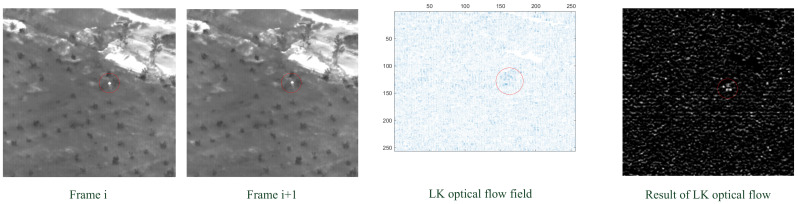
Infrared small moving target detection with the LK optical flow method.

**Figure 8 sensors-23-01660-f008:**
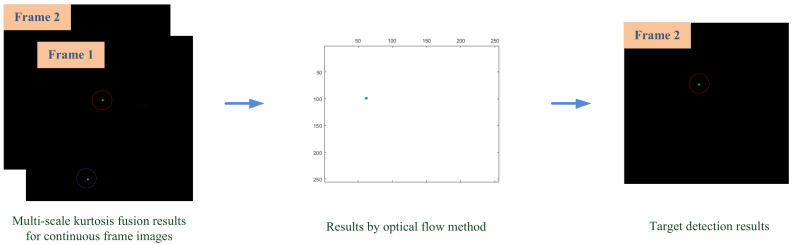
Target detection with the optical flow method for the output image sequence of kurtosis map fusion.

**Figure 9 sensors-23-01660-f009:**
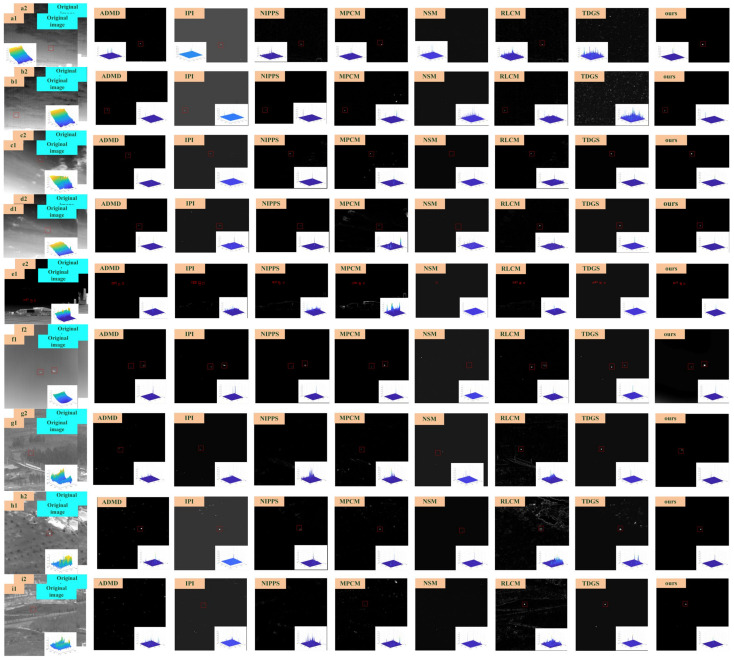
Target detection results. (**a**–**k**) in the figure is the infrared image of different targets in different backgrounds; 1,2 represents the pictures with the target motion changes in the continuous frames.)

**Figure 10 sensors-23-01660-f010:**
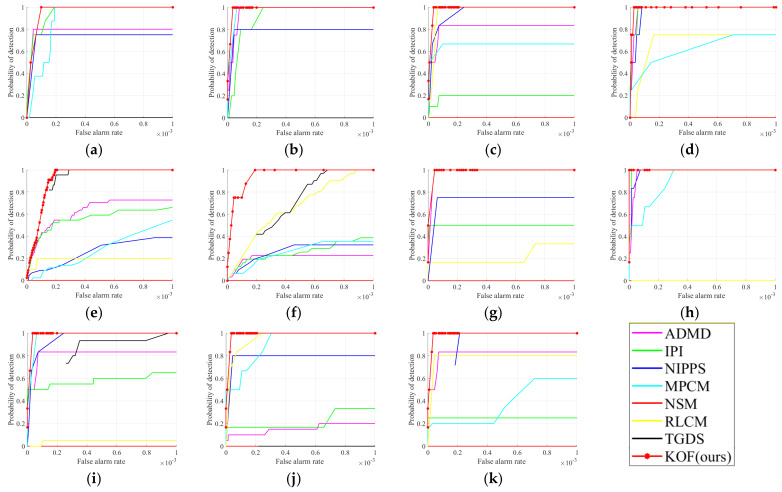
The ROC curves obtained by different methods. (**a**–**k**) corresponding to Figure 9a–k, respectively.

**Table 1 sensors-23-01660-t001:** Features of the datasets.

Sequence	Image Size	Target Description	Background Description
a	288 × 384	Low brightness, irregular shape	Sky background, highlight building edges, heavy noise, strong edges
b	288 × 384	Low brightness, irregular shape	Sky background, highlight building edges, heavy noise
c	288 × 384	High contrast compared with the background	Sky background, more cloud interference
d	288 × 384	High contrast compared with the background	Sky background, more cloud interference, strong edges
e	288 × 384	Multiple targets, target salient	Vehicle and building interference, bright background
f	256 × 256	Multiple targets, target salient	Overall background smoothing, heavy background noise
g	256 × 256	Dotted distribution, target dim	Highlighted background edge, trees interfere
h	256 × 256	Small circular target, target salient	Messy background, irregular bright background
i	256 × 256	Dotted distribution, target dim	Strong edges, too much noise
j	256 × 256	Dotted distribution, target dim	Irregular building shape, strong edges, heavy noise
k	256 × 256	Dotted distribution, target salient	Irregular building shape, strong edges, bright background

**Table 2 sensors-23-01660-t002:** SCRG values of different methods.

SCRG	ADMD	IPI	NIPPS	MPCM	NSM	RLCM	TGDS	Ours
a	27.36	33.86	26.78	19.70	22.71	18.55	4.27	39.11
b	32.91	33.35	31.73	17.89	25.07	10.27	13.73	34.73
c	36.24	38.35	39.60	26.53	17.15	19.22	38.21	42.82
d	29.72	35.80	26.71	15.26	27.66	27.18	32.21	34.80
e	36.97	27.52	40.32	35.29	10.04	53.35	52.96	53.83
f	34.29	36.56	37.21	37.90	24.40	27.14	29.83	40.09
g	39.21	33.91	23.71	5.14	27.77	30.16	37.04	42.34
h	21.67	24.17	27.52	12.64	23.87	6.10	23.29	32.53
i	29.53	32.24	23.07	21.90	12.74	14.85	27.76	32.37
j	32.19	33.46	36.54	27.07	21.50	28.33	35.17	41.81
k	42.32	43.28	47.39	29.14	13.21	45.98	47.86	52.63

**Table 3 sensors-23-01660-t003:** BSF values of different methods.

BSF	ADMD	IPI	NIPPS	MPCM	NSM	RLCM	TGDS	Ours
a	42.50	45.73	40.61	47.63	26.71	16.74	20.42	48.05
b	34.89	41.38	38.93	37.25	26.02	15.70	23.62	42.24
c	34.52	34.30	36.80	36.98	31.03	19.22	14.33	38.37
d	26.45	27.22	26.64	27.57	30.14	22.60	12.51	28.73
e	32.50	27.69	33.10	31.93	31.26	23.81	24.65	38.83
f	27.53	19.37	33.22	36.30	30.70	18.25	21.32	39.54
g	32.11	25.44	25.61	32.13	27.60	17.42	22.48	35.38
h	21.54	24.94	23.26	27.03	27.94	26.86	14.42	28.04
i	19.23	12.95	27.14	36.47	29.53	35.87	25.87	39.76
j	27.11	25.57	31.21	34.98	28.47	24.94	14.94	33.25
k	23.69	26.24	24.33	21.56	16.18	17.77	2.89	26.12

## Data Availability

The data supporting this achievement can be found in Data plane set of infrared dim small target detection and tracking underground/air background, at http://www.csdata.org/p/387/. This data is available from the public sector. The remaining data sets are laboratory data sets and are not published.

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
