# Peer review of "Infrared Small Target Detection Based on Multiscale Kurtosis Map Fusion and Optical Flow Method"

_sensors, 2023, doi:10.3390/s23031660_

Round 1

Reviewer 1 Report

This manuscript proposed a small IR target detection by combining multi-scale kurtosis map fusion (shape, size) and optical flow (motion) method.

The idea is very simple but the result is quite interesting.

To improve the paper, you have to verify the effect of each module (Kurtosis map fusion, optical flow).

For example, what if the target is remote and almost stationary in optical flow?

In noisy background, small targets are not easy to track in optical flow.

You have to test your method in low SCR, strong cluttered images.

In addition, deep learning part should be mentioned in introduction part.

For example,

AVILNet: A New Pliable Network with a Novel Metric for Small-Object Segmentation and Detection in Infrared Images  

Typos:

12. first, it is proposed a structure of infrared small target detection

198. is proposed to highlighted the target

eq. (4): line space in upper/lower position

Algorithm 2: line space in upper/lower position

Reviewer 2 Report

This paper proposed a weak target detection algorithm, which includes a multi-scale kurtosis map fusion method and an optical flow method. The experimental results show that the proposed method is effective. However, there are several questions to further improve the manuscript.

1. In the introduction, there is no discussion about the detection methods based on deep-learning and low-rank sparse.

2. Please check if there are any errors for the representative algorithms of DBT algorithm and TBD algorithm in the manuscript.

3. The number of the formula is missing, such as in line 259.

4. Please check the expression of the Algorithm 1 to ensure it is right.

5. Missing necessary references. For example, in line 364, the algorithms lack necessary citation.

6. The proposed method can detect the target compared with some old methods, however, there have been many related methods in recent years, and this paper did not analyze them and compared them in the experimental part.

7. Some pictures in this paper have low definition, such as Figure 9 and Figure 10, so the image quality should be improved.

8. Check the evaluation indicators in Table 3 to ensure it is right.

9. In experimental part, missing the necessary description. Therefore, the advantage of the proposed method can not be shown effectively.

Round 2

Reviewer 1 Report

All points were cleared in the final revision.

Reviewer 2 Report

Please check the original image in Figure 9,

Please check why there is "SCRG" in Table 3.
